# Assessment of Cervical Skeletal Trauma: The Synergistic Contribution of Forensic and Clinical Medicine to a Case of Corpse Concealment

**DOI:** 10.3390/healthcare11040510

**Published:** 2023-02-09

**Authors:** Ilaria Tarozzi, Lorenzo Franceschetti, Valentina Bugelli

**Affiliations:** 1Azienda USL Modena, U.O.C. Medicina Legale e Risk Management, 41126 Modena, Italy; 2Istituto di Medicina Legale, Dipartimento di Scienze Biomediche per la Salute, Università degli Studi di Milano, 20122 Milano, Italy; 3LABANOF, Laboratorio di Antropologia e Odontologia Forense, Istituto di Medicina Legale, Dipartimento di Scienze Biomediche per la Salute, Università degli Studi di Milano, 20122 Milano, Italy; 4Azienda USL TOSCANA SUD EST, U.O. Medicina Legale, 58100 Grosseto, Italy

**Keywords:** neck injuries, forensic anthropology, skeletonized, multidisciplinary approach

## Abstract

Fatal neck injuries represent a major challenge in forensic pathology because the anatomical complexity and high variability of neck structures make it often extremely difficult to differentiate true pathological findings from artifacts at autopsy. This topic becomes even more relevant when the forensic pathologist is required to make a pathophysiological evaluation of bone fractures in the absence of soft tissue to support the diagnosis. We report a case of unidentified, stone-covered, skeletonized human remains found within a pit below an abandoned building with bony lesions of the cervical spine and ribs, including a full-thickness fracture of the right lateral mass of the atlas (C1). After a careful study of the fractures was carried out by screening forensic literature and anthropological studies, clinical neurosurgical expertise was called upon to provide a reliable explanation. A rapid and violent twisting of the neck in the opposite direction from the fracture site by an attacker who pinned the victim’s torso is the scenario that most likely occurred in our case. This case report shows that the diagnosis of cervical spine injuries in skeletal remains should be the result of a multidisciplinary approach that integrates forensic, anthropological, and clinical expertise.

## 1. Introduction

Fatal neck injuries represent a major interpretative challenge for forensic pathologists, primarily because the anatomical complexity and high variability of neck structures make it often extremely difficult to differentiate true pathological findings from artifacts at autopsy [1,2]. Cervical spine lesions, in particular, are commonly found in traumatic deaths such as traffic accidents and fall-related blunt trauma, where fractures of cervical vertebrae are largely detected. Upper cervical spine (occipit-C2) fractures commonly represent the direct cause of death, while lower vertebrae (C3–C7) involvement is less often fatal [3,4].

Cervical spine fractures are a relatively little addressed theme in forensic literature except for discussions about specific types of injuries from mechanical energy, emphasizing the crucial diagnostic role of post-mortem radiology together with a careful inspection at autopsy [5,6,7,8,9,10,11,12,13]. Moreover, the systematic discussion about cervical fractures is mainly focused on their pathological consequences according to anatomical localization rather than pathophysiology; thus, forensic pathologists must necessarily resort to branches such as neurosurgery and traumatology with the aim of making a correct differential diagnosis of such a frequent autoptic finding. This topic becomes even more relevant in cases of skeletonized corpses, since the absence of soft tissues makes the pathophysiological interpretation of cervical fractures particularly difficult.

We report a case of unidentified skeletonized human remains presenting bone lesions that proved to be especially hard to interpret in terms of causation mechanism and dynamics as well as the evaluation of their contributory role to death. This case report illustrates how the determination of cause and manner of death in cases where cervical fractures are present requires proper integration between forensic and clinical sciences, especially if poor or no soft tissues are found.

## 2. Case Report

In the garden of a crumbling ancient villa located in an abandoned but accessible country place not far from the town, human remains were discovered lying inside a pit about 2 m deep. The pit was surrounded by thick vegetation and largely occupied by rubble, stones, soil residues, branches as well as roots belonging to adjacent plants. About 2–3 m above the pit there was a window which was accessible from inside the villa. The primary inspection demonstrated that the remains were almost completely skeletonized, mostly disarticulated and largely colonized by larvae of *Diptera* and *Coleoptera*.

The human remains were in prone position with the face in contact with the soil, left upper limb in extended and abducted position with clenched hand, right upper limb in flexed position with the forearm above the thorax, left lower limb slightly flexed at the knee with the foot facing laterally, and right lower limb in an extended position with the foot facing medially. Neither clothing nor personal effects were found. Other bone segments (the axis and a phalanx) were detected by further site inspection. 

A proper skeletal reconstruction was carried out before further examination of the skeletal remains. The head was almost completely skeletonized with a scarce scalp with dark straight hair. Cranial sutures were present and well differentiable. The following parameters were measured: skull height—19 cm; skull width—15.5 cm; base–bregmatic diameter—14.5 cm; nose–chin diameter—12.5 cm; bi-zygomatic width—13 cm; nose height—3 cm; nose width—3 cm; palate width—4 cm; palate length—5 cm. The lower jaw was disarticulated from the head with poor soft tissue residue. The dental formula was 13 14 15 16 17 18 25 26 27 28 (upper dental arch) and 45 46 47 48 32 33 34 35 37 (lower dental arch); in 36 and 38 there were closed dental alveolus. 

The neck and trunk were completely skeletonized. Cervical vertebrae, dorsal vertebrae from D1 to D11, clavicles, sternum, II-III-IV-V-VI-VII-VIII-XI-XII left ribs and I-II-III-IV-V-VI-XI-XII right ribs were disarticulated. The hyoid bone was absent. Anatomical relationships between D12, lumbosacral vertebrae and pelvis were preserved, with scarce soft tissue residue adherent to bone segments. The pubic symphysis presented a subpubic acute angle. The pelvis diameter was 13 × 12 cm. The left upper limb was disarticulated from the trunk, with scarce soft tissue residue adherent to the forearm and recognizable hand; no fingernails were present. The right upper limb was completely disarticulated and skeletonized. The humerus length was 36 cm; radius length was 26 cm; ulna length was 28 cm. The lower extremities were partially skeletonized with soft tissue residue to the joint with the pelvis and from the middle third of the thighs to the feet. The femur presented a length of 49 cm and head diameter of 4.8 × 4.7 cm. The tibia was 37.5 cm long. The fibula length was 38 cm. 

The remains presented a full-thickness fracture of the right lateral mass of C1 (in a radial pattern with irregular margins in the upper articular facet, obliquely linear with a posterior-medial trend and irregular margins in the lower articular facet), gouge of the spinous process of cervical vertebrae from C3 to C5, full-thickness fracture of the base of the C6 spinous process (arched and transversal, with irregular margins), not-full-thickness fracture of the I left rib anterior arch (linear, perpendicular, with irregular margins, 1.5 cm length) and full-thickness fracture of the II left rib posterior arch (obliquely linear with anterior-medial trend and irregular margins). 

The vertebral findings were further examined with a neurosurgeon expert in cranial and spine traumatology. A deep study of neurosurgical literature about vertebral trauma followed, with a subsequent comparison between different known causative mechanisms of injury and the respective lesion pattern. 

For a proper estimation of the time of death, the colonization time derived from the entomological examination was integrated with a post-mortem interval estimated by the application of the Total Body Score (TBS) and calculation of the Accumulated Degree Days (ADD) [14,15,16] referred to the average environmental temperature recorded by a data logger Escort Imini Plus placed in the discovery site. The minimum post-mortem interval (minPMI) was finally estimated at 35–45 days before discovery. 

Circumstantial data and anthropological measurements suggested that the victim could be a man who had disappeared 45 days before. A DNA test confirmed his identity. The DNA was extracted from a nail sample using the BioRobot EZ1 Advanced workstation with the Investigator Card and EZ1^®^DNA Investigator Kit (QIAGEN, Hilden, Germany). QuantStudio™ 6 Flex Real-Time PCR System (Thermo Fisher Scientific) and Quantifiler™ Trio DNA Quantification Kit (Life Technologies) were applied. DNA amplification was performed applying the PowerPlex^®^ Fusion 6C System (Promega, USA). DNA was sequenced using the 3500 Genetic Analyzer (Thermo Fisher Scientific) and analyzed by GeneMapper™ ID-X Software.

The remains were finally identified as being those of a reported missing 33-year-old Chinese man who was suspected to be a victim of a mafia murder.

The death was attributed to fatal spraining and cervical trauma. The manner of death was determined to be homicidal.

## 3. Discussion

The accurate interpretation of the cause and manner of death represents a major challenge for forensic pathologists in general. In the case of badly preserved or even skeletonized corpses, however, it can prove to be particularly difficult because of the more or less complete absence of soft tissues, which in well-conserved cadavers provide fundamental information about the nature, entity and contributory role to the death of eventual traumatic injuries [17,18]. Moreover, not all lesions clearly visible in soft tissues reach the underlying bone; thus, the only presence of skeletal segments may lead to the underestimation of traumatic injury load. On the other hand, the correct interpretation of bone lesions may suffer from modifications induced by time and taphonomic factors as well; thus, their nature and production period (peri- or postmortem) can hardly be defined. Finally, smaller fractures can be difficult to reveal at autopsy, resulting in an incomplete lesion pattern detection [17]. Further difficulties can derive from the reconstruction of suspected criminal events when the site of discovery is represented by a complex scenario, such an eventuality requiring a particularly articulated inspection including the proper detection, recording, and recovering of environmental elements [19]. This is an essential step to obtain useful information about the potential crime scene which could be integrated with frequently poor circumstantial data in order to reach a reliable reconstruction of the cause and manner of death.

In decomposing soft tissues, trauma or signs of injury such as ecchymoses and hematomas are not always visualized [20]. In challenging cases of decomposed corpses, the forensic analysis can therefore benefit from the application of post-mortem radiology and scanning electron microscopy with energy-dispersive X-ray use (SEM/EDX). These instrumental procedures can play an important part in synergy with the forensic pathologist’s examination. 

Post-mortem radiology has a relevant role mainly in fresh cadavers in cases of firearms or stab wound injuries, for instance. It can be helpful in advanced decomposed corpses as well, however, when soft tissues are still present and the exclusion of bony trauma is essential. The application of the SEM/EDX analysis detects any exogenous material that may have been depositing on a body before death [19]. Through this analysis, the correlation between a suspected weapon and the wounds identified on the victim’s body can be established, or, in cases where no weapon is detected, information about the nature of the weapon used can be provided to the investigators. These instrumental analyses were not carried out within the scope of the investigations.

In the presented case, the remains were almost completely skeletonized and mostly disarticulated; they were lacking in some small bone segments, including the hyoid bone. Apart from a scratch on the spinous processes of all vertebrae from C3 to C5, the cervical spine was affected by a full-thickness fracture of the right lateral mass of C1 and of the base of the C6 spinous process; not-full-thickness fractures of the I left rib anterior arch and a full-thickness fracture of the II left rib posterior arch were also detected. The remaining skeletal structure was intact. Despite poor circumstantial information, it was the forensic inspection which initially provided the fundamental information to lead to interpretation about the cause and manner of death. As described above, the site of discovery was an abandoned villa that was in an isolated but accessible place. The remains were lying inside a pit about 2 m deep with no clothing or personal effects in close proximity. The major decomposition of body parts in contact with the soil suggested that the victim remained in a prone position until discovery. A homicidal manner of death with corpse concealment was then hypothesized, the latter because of the presence of rubble and stones inside the pit not belonging to the surrounding environment and the absence of clothes near the remains. In order to provide evidence to further corroborate the investigative hypothesis about a murderous event, the study of bone lesions focused on the cervical vertebrae.

Cervical spine injuries represent a particularly hard challenge in forensic pathology even when cadavers are well-preserved. In the case of skeletal human remains, however, the lack of soft tissues makes it furtherly complicated to define cervical lesions’ origin and contributory role to death just because of the wide variety of possible causative mechanisms. According to the affected cervical vertebrae, lesions’ characteristics and the presence of bone injuries involving other bones than the cervical spine, in fact, a different etiopathogenesis can be proposed.

The major contributory role of the upper cervical spine’s fractures in fatal cases is easily explained by their most frequent association with skull and brain injuries as well as the increased spinal cord’s susceptibility to hyperextension/hyperflexion injury typically affecting the upper two cervical vertebrae (that provide most of the rotational movement of the head rather than neck flexion and extension) [3,21]. Atlas (C1) fractures can affect the posterior arch, in the case of hyperextension and axial loading; the anterior arch, mainly due to hyperextension; or the lateral mass, as a result of the asymmetric lateral compression forces during distraction injuries. Atlas burst (Jefferson) fractures are secondary to axial loading and involve both anterior and posterior arches [22]. Axis (C2) fractures more frequently involve the odontoid process or the neural arch between superior and inferior articular processes, in the latter case determining the so-called “Hangman’s fracture”, which is the bilateral fracture of the second cervical vertebra with an anterior dislocation of its body, with the dens remaining intact as a result of a violent and rapid hyperextension-distraction of the neck [13]. Among rostral cervical lesions there is also the atlanto-occipital dislocation, resulting from craniocervical ligament separation, and atlanto-axial dislocations, secondary to the rupture of the transverse ligament [4,23]. Instead, occipital condyle fractures constitute more clinical rather than autoptical findings because of their good outcomes in the absence of other associated injuries [23]. Because of the presence of a window which was located about 2–3 m above the pit and accessible from inside the villa, a fall from a low height was at first taken into account. Skeletal trauma resulting from a fall can involve different body regions, mainly depending on the primary impact (Figure 1): in the case of falls on the head, a massive skull fracture is typically present, together with an arm fracture in the case of a defense attempt during falls (Figure 1A,B); falls on to the feet commonly involve lower limbs and the pelvic girdle (Figure 1C); the axial skeleton can be broken in different points depending on whether the body falls on the head, the feet or the side. In our case, however, bone fractures were exclusively distributed among the cervical spine and the rib cage (Figure 1D), so the hypothesis of a fall was excluded. 

No entry or exit wounds were found as well as bone injuries clearly attributable to shotgun wounds. Moreover, since all fractures presented irregular margins, stab wounds were excluded as well. Considering the discovery of rubble and stones inside the pit, blunt trauma was further considered. If the body hits or is hit by a blunt surface, in fact, the localized absorption of a critical amount of mechanical energy determines a structural alteration of the anatomic integrity of the affected site. The extent of injury depends on the amount of force exerted [3]. In this case, C6 and the I-II left ribs presented linear mostly full-thickness fractures with irregular margins which could have been determined by the localized application of blunt forces such as those exerted by heavy stones thrown with considerable force (lapidation) and/or fallen from a height. The scratch of the spinous process of vertebrae from C3 to C5 was consistent with a blunt causative mechanism as well; however, its non-specific pattern made it attributable to macrofauna action as well. Regarding the first cervical vertebra, it presented an isolated fracture of a lateral mass with no involvement of the adjacent bone. Considering its deep anatomic location, however, it was less likely that direct/indirect blunt trauma occurred, sparing the skull and the axis.

Proceeding with the analysis of the mechanism of trauma that occurred, the authors realized that there was an absence of data in the relevant forensic literature that would best justify a C1 injury. Therefore, research was conducted on different fronts, in the more general field of clinical medicine, focusing on those branches dealing with trauma to the cervical spine, i.e., neurosurgery and traumatology. Neurosurgical literature provides an interesting explanation: lateral mass fractures of C1 are caused by asymmetric lateral compression forces during distraction injuries [23,24]. A rapid and violent neck torsion (in the opposite direction of the fracture site) by an assailant blocking the victim’s torso was the scenario that most likely happened in our case (Figure 2). 

The case was then brought to the attention of an expert neurosurgeon, who provided fundamental information for a better understanding of C1 injury pathophysiology. According to the clinical long-standing expertise, fractures of C1 are considered unique compared to other vertebrae because of the peculiar anatomic characteristics of the atlas. It is the most cranial component of the cervical spine, with a ring-shaped form and direct articulation with the occiput and axis through a ligamentous apparatus which provides stability to the upper cervical spine. Although C1 fractures are generally considered at a low-risk of severe sequelae, unilateral or comminuted fractures may be unstable, causing the displacement of the proximal bone structures. 

No topographic proximity to vital anatomic structures exists with regard to vertebrae from C3 to C6 and the I left rib as well, while the II left rib fracture may have potentially involved the lungs, pleura and vessels. On the other hand, the C1 fracture by application of asymmetric lateral distraction forces may have been associated to the vasculo-nervous neck structures injury. In particular, the compression or stretching of carotid arteries including the carotid sinus, laryngo-tracheal axis and vagus nerve trunk may impair cardiocirculatory, respiratory and brain function as well. The authors therefore concluded that the fracture of the first cervical vertebra played the main contributory role to the death. 

In well-conserved cadavers, cervical spine injuries are commonly interpreted from their productive mechanism’s point of view. In the case of hanging, for example, circumstantial data suggests that a long drop has been used, and a forensic pathologist looks for cervical fractures. However, skeletal remains are devoid of soft tissues that usually lead to the etiopathological interpretation of traumatic injuries; thus, the study of lesions patterns represents the only possible diagnostic approach.

As mentioned above, however, the forensic literature is poor in the systematic treatment of cervical fractures according to their etiopathogenesis; thus, a clinical contribution is necessarily required. 

## 4. Conclusions

The interpretation of cervical spine injuries is particularly difficult in the case of skeletonized bodies because of the lack of valuable information from soft tissues. Since no etiopathological hypothesis can then be formulated if not suggested by circumstantial data, a differential diagnosis of cervical fractures is necessarily based on lesional characteristics. That is why forensic pathologists often need to consult clinicians to understand which causative mechanism may have such a traumatic injury pattern. The interpretation of cervical spine injuries in skeletal remains should result from a multidisciplinary approach integrating forensic, anthropological, and clinical expertise. 

## Figures and Tables

**Figure 1 healthcare-11-00510-f001:**
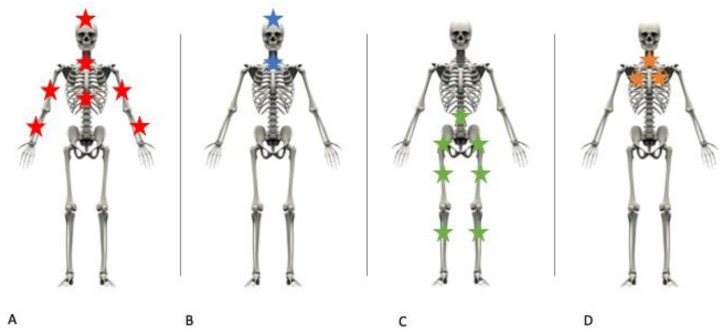
Lesions’ distribution in case of falls onto the head with (**A**) and without (**B**) defense attempt, falls on to the feet (**C**) and in our case (**D**).

**Figure 2 healthcare-11-00510-f002:**
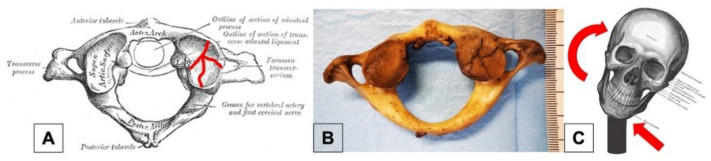
Right lateral mass fracture of C1 (**A**) in our case (**B**); schematic illustration of productive mechanism (**C**). Pictures A and C were adapted from Anatomy of the Human Body by Henry Gray, PHILADELPHIA: LEA & FEBIGER, 1918.

## Data Availability

Data presented in this study are available on request from the corresponding author. The data are not publicly available to privacy restriction.

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
