# Peer review of "Assessment of Cervical Skeletal Trauma: The Synergistic Contribution of Forensic and Clinical Medicine to a Case of Corpse Concealment"

_healthcare, 2023, doi:10.3390/healthcare11040510_

Round 1

Reviewer 1 Report

The authors presented a case report on a body concealment. The forensic practitioners face the case in an appropriate way providing the authorities good basis to solve the homicide.

The importance of a comprehensive approach, both from a forensic and a clinical way is an important topic, and excellent considerations of key factors about trauma analyses were discussed. Implications of the findings are impactful to improving the multidisciplinary approach in such scenarios. The authors should discuss more -even if it is not pivotal for the presented case- the role of forensic radiology. This recommendation is intended to clarify an important point in the paper and strengthen the contribution of this case report to the literature.

Author Response

Dear Editor,

Thank you very much for your kind email and the reviews included in your response.

We are glad to inform You that quite all the criticisms raised by the Reviewers have been accepted.

Please, find our reply to Reviewer’s comments as follows:

Response to Reviewer #1:

R1: The authors presented a case report on a body concealment. The forensic practitioners face the case in an appropriate way providing the authorities good basis to solve the homicide. The importance of a comprehensive approach, both from a forensic and a clinical way is an important topic, and excellent considerations of key factors about trauma analyses were discussed. Implications of the findings are impactful to improving the multidisciplinary approach in such scenarios. The authors should discuss more -even if it is not pivotal for the presented case- the role of forensic radiology. This recommendation is intended to clarify an important point in the paper and strengthen the contribution of this case report to the literature.

A1: Thank you very much for this comment. We briefly integrated the suggestion as follow:

In challenging cases of decomposed corpses, the forensic analysis can therefore benefit from the application of post-mortem radiology and scanning electron microscopy with energy-dispersive X-ray Use (SEM/EDX). These instrumental procedures that can play an important part in synergy with the forensic pathologist’s examination. 

Post-mortem radiology has a relevant role mainly in fresh cadavers in cases of firearms or stab wounds injuries, for instance. It can be helpful in advanced decomposed corpses as well, however, when soft tissues are still present and exclusion of bony trauma is essential. The application of the SEM/EDX analysis detects any exogenous material that may have been depositing on a body before death [19]. Through this analysis, the correlation between a suspected weapon and the wounds identified on the victim’s body can be established, or, in cases where no weapon is detected, information about the nature of the weapon used can be provided to the investigators. These instrumental analyses were not carried out within the scope of the investigations.

We have really appreciated the improvement of the manuscript due to the implementation of the comments.

Sincerely,

The Corresponding Author

Reviewer 2 Report

The paper entitled "The assessment of cervical skeletal trauma: the synergistic contribution of forensic and clinical medicine to a case of corpse concealment." is very interesting. This paper deals with a particular case of homicide with concealment. The article is well written and only minor spelling English language corrections are required.  The main task of this paper is to create a tight connection between forensic practice and clinical medicine.  I suggest to the authors should discuss the use of scanning electron microscopy. Such guidance is believed to increase the quality of the discussion.  

Author Response

Dear Editor,

Thank you very much for your kind email and the reviews included in your response.

We are glad to inform You that quite all the criticisms raised by the Reviewers have been accepted.

Please, find our reply to Reviewer’s comments as follows:

Response to Reviewer #2:

R1: The paper entitled "The assessment of cervical skeletal trauma: the synergistic contribution of forensic and clinical medicine to a case of corpse concealment." is very interesting. This paper deals with a particular case of homicide with concealment. The article is well written and only minor spelling English language corrections are required.  The main task of this paper is to create a tight connection between forensic practice and clinical medicine.  I suggest to the authors should discuss the use of scanning electron microscopy. Such guidance is believed to increase the quality of the discussion.

A2: Thank you very much for this comment. We briefly integrated the suggestion as follow:

In challenging cases of decomposed corpses, the forensic analysis can therefore benefit from the application of post-mortem radiology and scanning electron microscopy with energy-dispersive X-ray Use (SEM/EDX). These instrumental procedures that can play an important part in synergy with the forensic pathologist’s examination. 

Post-mortem radiology has a relevant role mainly in fresh cadavers in cases of firearms or stab wounds injuries, for instance. It can be helpful in advanced decomposed corpses as well, however, when soft tissues are still present and exclusion of bony trauma is essential. The application of the SEM/EDX analysis detects any exogenous material that may have been depositing on a body before death [19]. Through this analysis, the correlation between a suspected weapon and the wounds identified on the victim’s body can be established, or, in cases where no weapon is detected, information about the nature of the weapon used can be provided to the investigators. These instrumental analyses were not carried out within the scope of the investigations.

We have really appreciated the improvement of the manuscript due to the implementation of the comments.

Sincerely,

The Corresponding Author

Reviewer 3 Report

·       Line 4: Is a dot / full stop needed at the end of the ‘title’?

·       Line 28: The multidisciplinary approach (consultation with clinicians) is not apparent in the ‘case report’ section.

·       Introduction: End the first paragraph with the following sentence [Upper cervical spine (occipit-C2) fractures commonly represent the direct cause of death, while lower vertebrae (C3-C7) involvement is less often fatal]. Accordingly, cite the relevant references at the end of this sentence.

·       Introduction: Begin the 2nd paragraph with the following sentence (Cervical spine fractures are poorly treated in forensic literature if not in the context of discussions about specific types of injuries from mechanical energy, frequently emphasizing the crucial diagnostic role of post-mortem radiology together with careful inspection at autopsy [5-13].) and combine this sentence with the next 2 paragraphs of the ‘introduction’ section.

·       Introduction: Combine the present last 2 paragraphs together. Therefore, accordingly, the revised manuscript will consist of only 3 paragraphs.

·       Line 39 (Introduction): Replace “poorly treated” with a more appropriate term/phrase/word.

·       Lines 45-46 (Introduction): Mention the other specialist branches (mention the other relevant specialist branches).

·       Line 45 (Introduction): Replace “physiopathology” with “pathophysiology”.

·       Lines 52-53 (Introduction) – “production mechanism and dynamics”: I understand what you are trying to refer to here. However, it would be better to avoid “production” and instead the term “causation” appropriately while referring to the dynamics and mechanism of causation of injury (or bone lesions as mentioned in the former half of the sentence).

·       Line 53 (Introduction) – “definition of their contributory role to death”: Avoid the word “definition” and accordingly rewrite the sentence without any change in the intended meaning.

·       Line 54 (introduction): In addition to “bodies”, mention “cervical skeletal remains”.

·       Introduction-Last sentence: This sentence should be rephrased

·       Case report-1st paragraph: Indicate that this is a case report from Italy.

·       Line 66: Replace “they” with “The human remains”.

·       Line 72: Replace “post-mortem examination” with “further examination of the skeletal remains”.

·       Line 73: Avoid using the word “residue” and accordingly rewrite the sentence.

·       Line 75 – “skull length”: Are you referring to “skull height”?

·       Line 77 – “jaw”: Are you referring to the lower jaw/mandible?

·       Lines 78-80 – “dental formula”: It would be better to consider the FDI (Federation Dentaire Internationale) notation / FDI dental numbering system. If you do not prefer to change the dental notation, then provide details of the dental formula used to note the teeth.

·       Line 105 – “from discovery”: Should it be “before discovery”?

·       Line 107: Provide details about the DNA test leading to the identity of the human remains. Specimens collected for DNA analysis? Reference samples used for comparison? And so on.

·       Line 110 – “cardiorespiratory arrest”: This phrase must NOT be used while referring to the cause of death.

·       Case report: The ‘introduction’ section refers to the importance of consultancy with clinicians. However, there are no related details mentioned in the ‘case report’ section. Please note that your objective of publishing this case report (as depicted in the ‘abstract’ and ‘introduction’ sections and even the ‘title’ of the manuscript) is not reflected in the ‘case report’ section. The role played by the other specialists in the inference of cervical skeletal remains examination is not explained in the ‘case report’ section.

·       Line 124 – “in front of a”: Avoid these words and accordingly rewrite the meaning without a change in the intended meaning.

·       Line 137: Replace “judicial” with another word that means the same (to avoid any legal inference from the word “judicial” that is used).

·       Lines 146-149: Avoid single-sentence individual paragraphs.

·       Case report: Few aspects of the present case are mentioned for the first time in the ‘discussion’ section. All information about the present case should first be mentioned in the ‘case report’ section before being discussed in the ‘discussion’ section.

·       Figure 1: The depiction of bone lesions / skeletal lesions is excellent. The authors should confirm in their response to reviewers’ comments that the figure is their original figure.

·       Figure 2: The authors should confirm in their response to reviewers’ comments that the figure is their original figure. If not, relevant copyright-related permission to reproduce the pictures should be obtained and evidence provided.

·       Figures 1 and 2 – legend/title/caption: Begin the first word with a capital alphabet/letter.

·       Lines 184-205: Avoid very small paragraphs; club them appropriately.

·       Line 184: Delete “including”.

·       Lines 188-190: Discuss the relevance of this sentence with reference to the findings in the present case. After reading the next paragraph, it is apparent that this is explained in the new paragraph. Therefore, it is important to club paragraphs to avoid the loss of flow of thoughts.

·       Lines 191-195: The 2nd sentence should be rewritten as it lacks clarity.

·       Line 197: Replace “productive” with “causative”.

·       Line 208: And so?

·       Lines 208 and 227: Replace “Literature” with “literature”.

·       Lines 216-220: Explain further as to how cardiorespiratory arrest would occur in cases of those injuries listed by the authors in the preceding sentence.

·       Discussion – last sentence (last paragraph) + the paragraph before Figure 1: Going through these paragraphs, gives me the impression that the authors (forensic practitioners) did not consult other specialists/clinicians (neurosurgeons/traumatologists), but basically referred to clinical literature in concluding the authors (forensic practitioners) inference about the causation of the cervical injuries. However, in the ‘introduction’ section and elsewhere (title/abstract/conclusion) they provide an impression that other specialists were consulted.

·       Lines 231-240 (Conclusions): Avoid multiple single sentence paragraphs; provide a single paragraph to represent the section on conclusions.

·       Line 237: Delete “productive” or replace “productive mechanism” with “causative mechanism”. I prefer to replace “productive mechanism” with “causative mechanism”.

·       Line 239: Replace “Diagnosis” with “Interpretation”.

·       Lines 248 and 251: Replace “not” with “Not” & replace "the" with "The", respectively.

Author Response

Dear Editor,

Thank you very much for your kind email and the reviews included in your response.

We are glad to inform You that quite all the criticisms raised by the Reviewers have been accepted.

Please, find our reply to Reviewer’s comments as follows:

Response to Reviewer #3:

Thank you very much for comments. In this reply we do not report line by line the reviewers suggestions that we have entirely integrated in the manuscript.

Regarding the lack of clarity about having consulting another specialist, we have provided furterh explanation both in “case report section” and “discussion section”.

“..Vertebral findings were further examined with a neurosurgeon expert in cranial and spine traumatology. A deep study of neurosurgical literature about vertebral trauma followed, with subsequent comparison between different known causative mechanisms of injury and respective lesion pattern..”.  Proceeding with the analysis of the mechanism of trauma that occurred, the Authors realized that there was an absence of data in the relevant forensic literature that would best justify C1's injury. Therefore, research was conducted on different fronts, in the more general field of clinical medicine, focusing on those branches dealing with trauma to the cervical spine, i.e., neurosurgery and traumatology.  Neurosurgical literature provides an interesting explanation: lateral mass fractures of C1 are caused by asymmetric lateral compression forces during distraction injuries [23]. A rapid and violent neck torsion (in the opposite direction of the fracture site) by an assailant blocking the victim's torso was the scenario that most likely happened in our case (fig.2). The case was then brought to the attention of an expert neurosurgeon, who provided fundamental information for a better understanding of C1 injury pathophysiology. According to clinical long-standing expertise, fractures of C1 are considered unique compared to other vertebrae because of atlas peculiar anatomic characteristics. It is the most cranial component of the cervical spine, with ring-shaped form and direct articulation with the occipit and the axis through a ligamentous apparatus which provides stability to the upper cervical spine. Although C1 fractures are generally considered at low-risk of severe sequelae, unilateral or comminuted fractures may be unstable causing displacement of the proximal bone structures”.

We have really appreciated the improvement of the manuscript due to the implementation of this comment because the reviewer has pointed out our intent even if it was not so clearly detailed in the text.

Regarding figure 1 the it is authors original figure. Regarding figure 2b it is author original one, while figure 2a and c are adapted from Grey’s anatomy manual 1918.

Finally, we have not indicated that it is a case report from Italy in order to guarantee the anonymity of the case which is actually an open judicial case.

Sincerely,

The Corresponding Author

Reviewer 4 Report

I enjoyed reading your manuscript and only made a few suggested changes either for clarity or for my own interest. Your manuscript was interesting to read, relevant to the forensic community and in my opinion worthy of publication. Thank you for your interesting paper and all your hard work! 

Author Response

Dear Editor,

Thank you very much for your kind email and the reviews included in your response.

We are glad to inform You that quite all the criticisms raised by the Reviewers have been accepted.

Please, find our reply to Reviewer’s comments as follows:

Response to Reviewer #4:

R4: I enjoyed reading your manuscript and only made a few suggested changes either for clarity or for my own interest. Your manuscript was interesting to read, relevant to the forensic community and in my opinion worthy of publication. Thank you for your interesting paper and all your hard work! 

A4: Thank you very much for this comment. We are glad not only that the reviewer has appreciated the topic of our manuscript but also the challenging points for forensic pathologists. About suggestions authors have inserted the changements and improved the English language. Please check it.

We have really appreciated the improvement of the manuscript due to the implementation of the comments.

Sincerely,

The Corresponding Author
